# Dynamics of Serum Thymidine Kinase 1 at the First Cycle of Neoadjuvant Chemotherapy Predicts Outcome of Disease in Estrogen-Receptor-Positive Breast Cancer

**DOI:** 10.3390/cancers13215442

**Published:** 2021-10-29

**Authors:** Bernhard Tribukait

**Affiliations:** 1Department of Oncology-Pathology, Karolinska Institute and University Hospital Solna, 17164 Stockholm, Sweden; bernhard.tribukait@ki.se; 2Cancer Centrum Karolinska, CCK, Plan 00, Visionsgatan 56, Karolinska Universitetssjukhuset, Solna, 17164 Stockholm, Sweden

**Keywords:** circulating thymidine kinase 1, cell loss, biomarker, early treatment response, breast cancer

## Abstract

**Simple Summary:**

Chemotherapy before surgery (NAC) is an option for high-risk breast cancer (BC) patients. Pathologic complete response (pCR) predicts long-term outcome and has become a surrogate biomarker for survival. pCR is, however, reached in only <10% of hormone-receptor-positive (ER+) patients and is of limited prognostic value. Biomarkers able to predict outcome early during NAC would facilitate individualized therapy with the possibility to adjust or interrupt an ineffective therapy. Here, it is shown that differential response of the serum concentration of thymidine kinase 1, an enzyme involved in the DNA synthesis and released from the tumor into the blood, 48 h after the first cycle of NAC, predicts long-term outcome in localized advanced ER+/HER2-BC. The different reactions to chemotherapy could be used to guide this process early during NAC and utilized to identify mechanisms of tumor sensitivity that could provide a prediction of long-term outcome prior to chemotherapy.

**Abstract:**

Pathologic complete response (pCR) predicts the long-term outcome of neoadjuvantly treated (NAC) breast cancer (BC) but is reached in <10% of hormone-receptor-positive patients. Biomarkers enabling adjustment or interruption of an ineffective therapy are desired. Here, we evaluated whether changes in the serum concentration of thymidine kinase 1 (sTK1) during NAC could be utilized as a biomarker. In the PROMIX trial, women with localized HER2- BC received neoadjuvant epirubicin/docetaxel in six cycles. sTK1 was measured with an ELISA in 54 patients at cycles 1–4 and in an additional 77 patients before and 48 h after treatment 1. Treatment resulted in a 2-fold increase of sTK1 before and a 3-fold increase 48 h after the cycles, except for the first cycle, where half of the patients reacted with a significant decrease and the other half with an increase of sTK1. In Kaplan–Meier estimates of ER+ patients divided by the median of the post/pre-treatment sTK1 ratio at the first treatment cycle, OS was 97.7% and 78% (*p* = 0.005), and DFS was 90.7% and 68% (*p* = 0.006), respectively. Thus, the response of sTK1 at the first cycle of chemotherapy could be used both as an early biomarker for the guidance of chemotherapy and for the study of inherent tumor chemo-sensitivity, which could predict long-term outcome prior to therapy.

## 1. Introduction

Neoadjuvant chemotherapy (NAC) is an increasingly used therapeutic option, particularly in younger women with locally advanced, high-grade, estrogen-receptor-negative (ER-) breast cancer (BC) and more advanced nodal disease [1,2]. NAC increases the rate of breast preservation [2], pathologic complete response (pCR), however, varies considerably between various subgroups, with 8% in hormone-receptor-positive (HR+)/HER2-tumors and 39% in hormone-receptor-negative (HR-)/HER2+ subsets [3]. For the translation of the increasing body of knowledge of tumor biology into clinical practice, NAC is important due to the possibility of the direct observation of the tumor [4]. However, in an individual case, the response to NAC, as reflected in tumor volume at surgery, is difficult to predict, and tumor properties determining the response to therapy apparently go beyond the characterization by routine pre-treatment analyses. The growing awareness of this problem has stimulated the search for biomarkers that can be used for early evaluation of tumor response, promoting individualized treatment. Identifying tumors with poor response during treatment would enable ineffective therapy to be adjusted or interrupted. Because the time frame to modify an ineffective therapy becomes smaller with the number of treatment cycles, poor response to treatment should be established as early as possible.

Methods for evaluating tumor response over the course of NAC comprise (I) measurement of tumor volume, (II) estimation of tumor metabolism via its uptake of radioactive tracers, and (III) measurement of the levels of macromolecules in the blood released from disrupted tumor cells. Response criteria for anatomical measurements, which are the most frequently used, have been defined in the Response Evaluation Criteria in Solid Tumors, RECIST [5]. However, routine clinical methods for estimation of tumor volume vary in accuracy, and there is a delay between tumor cell disruption and macroscopic shrinkage of the tumor. Molecular imaging, such as ^18^F-fluorodeoxyglucose PET/CT, combines measurements of size with those of tumor metabolism [6]. In BC, ^18^F-FDG PET/CT after two courses of NAC was able to predict pT0 in HER2+ and triple-negative tumors with sufficient accuracy but was less conclusive in the ER+/HER2- tumor subgroup [7]. Recently, textural evaluations on images acquired already prior to the start of NAC were used to predict the pathological response of the tumor [8]. The basic concept of such studies is that micro-structural tumor characteristics are associated with functional tumor properties, such as sensitivity to chemotherapy. With regard to the release of macromolecules from disrupted tumor cells into the blood, an essential issue is that chemotherapy often causes massive cell disruption in normal tissues with high proliferation and that the volume of these tissues may be much greater than that of the tumor.

When methods became available to identify tumor-derived elements in the blood, such as DNA fragments, nucleosomes, RNA and exosomes, concentration measurements over time also made it possible to assess response to therapy [9]. The excess of normal tissue-derived materials, however, remained as one of the difficulties in the quantitative assessment of genetic/epigenetic abnormalities in the blood.

For ER+/HER2- metastatic BC, which is the most common subtype, endocrine therapy combined with CDK4/6 inhibitors has become a standard treatment. Despite improvements in survival, a major challenge is the primary resistance or development of resistance. In search of possibilities to recognize resistance, various circulating biomarkers such as tumor DNA, RNA, exosomes, circulating tumor cells and thymidine kinase 1 (TK1) have been evaluated [10].

Elevated levels of TK1 in blood have been found in a variety of malignancies, including BC [11,12]. TK1 is a pyrimidine salvage enzyme that recycles deoxyribonucleotides from degraded DNA and catalyzes the phosphorylation of deoxythymidine to thymidine monophosphate, subsequently used in DNA synthesis and repair of DNA [13,14]. The cellular level of TK1 is cell-cycle-dependent; it is unmeasurable in G0/G1 but increases at the G1/S-phase border and reaches peak values during late S-phase/G2. At mitosis, it is degraded via ubiquitination [15,16]. Thus, cell proliferation does not normally result in any leakage of TK1 to the extracellular compartment. In serum from healthy human subjects, there is, nevertheless, a background level of about 0.24 ng/mL [17], probably connected with the clearance of non-viable normal cells but eventually reflecting normal tissue proliferation failures and non-apoptotic cell death.

In an earlier study, we explored the possibility to use the concentration of TK1 in blood as a marker of tumor cell loss to predict early during chemotherapy pathological tumor response [18]. Patients with advanced but localized BC received neoadjuvant docetaxel-based chemotherapy. As a measure of cell loss, sTK1 was related to the routinely determined tumor volume. This cell-loss metric, obtained in connection with the second cycle of therapy, was significantly associated with the pathologic outcome at surgery after six cycles of NAC.

The aim of the present study was to evaluate the prognostic significance of dynamics of sTK1 changes during NAC in patients with newly detected BC without distant metastases. Concentration changes were expressed as the ratio of sTK1 48 h post/pre-treatment. This measure, independent of tumor volume, would permit a more rigorous quantitative evaluation of tumor response than the earlier cell loss measurements. In 54 patients, complete sets of serum before and 48 h after treatment cycles 1–4 were available. Differences in response at treatment cycle 1 observed in the 54 patients could be confirmed in additional 77 patients. Possible associations of this metric with baseline patient and tumor characteristics were evaluated, as well as the prognostic significance by disease-free survival (DFS) and overall survival (OS) after 70 months of follow-up.

## 2. Results

In the flow chart (Appendix A), the number of patients where serum samples were available is accounted for. In 54 patients, all eight serum samples were present prior to and 48 h following treatment cycles 1–4 and at cycle 1 in 77 additional patients (Table 1).

The two groups were similar with respect to all patient and tumor baseline characteristics. They were, therefore, combined in the analysis of sTK1 response to treatment 1. The median age (range) of the 131 patients was 49.8 (30–70.6) years; the median tumor size of 55 (20–180) mm corresponded to a volume of 87 (4.2–3052) cm^3^, and according to the stage, 39% were T2 and 54% T3. Positive ER (ER+) and PR (PR+) status (≥10%) was present in 72% and 54%, respectively, and positive lymph nodes were found in 57%. The median (range) of the Ki67 labelling index was 30% (1–90%), and sTK1 was 0.31 (0.1–1.29) ng/mL. According to the St. Gallen surrogate definition of the intrinsic subtypes [19], 48 of the tumors were luminal-A-like, 48 luminal-B-like, and 35 TBNC. HER2-status was 0 or 1+ in 73% and 2+ in 27%.

In the 54 women where pre- and post-treatment values were available for cycles 1–4, the group means displayed a characteristic response pattern (Figure 1).

For cycle 1, the 48 h post-treatment value (median 0.28; range 0.13–1.12 ng/mL) was similar to that of the baseline (0.30; 0.10–0.72 ng/mL). For treatment 2, three weeks later, the pre-treatment median value was substantially greater (0.58; 0.15–2.65 ng/mL). Each treatment following cycles 2, 3, and 4 resulted in maximum post-treatment TK1 concentrations with a median of 0.76–0.82 ng/mL. During the rest periods between cycles 2–3 and 3–4, sTK1 decreased to pre-treatment levels corresponding to the level before cycle 2.

For each patient and treatment cycle, the sTK1 response was expressed in relative terms, i.e., as the ratio between the post/pre-treatment sTK1 value. Figure 2 shows the group means (*n* = 54) for this ratio at treatment cycles 1–4. The ratio was close to 1.0 at cycle 1; it was 1.5 at cycle 2 and appeared to reach a plateau value at approximately 2.0 at cycles 3 and 4.

Scrutiny of the individual data reveals that the apparent lack of response to treatment cycle 1 is due to the fact that approximately half of the patients responded with an increase in sTK1, whereas the other half showed a decrease. Based on the median of the sTK1 post/pre-treatment ratio at cycle 1, the patients were categorized into a low-ratio group (group A, *n* = 27; median 0.74, range 0.30–1.11) and a high-ratio group (group B, *n* = 27; median 1.38, range 1.12–2.53) (Figure 3, upper part).

As illustrated in Figure 3 (lower part), these groups differed significantly with respect to baselines for sTK1. The low-ratio group (A) had a baseline of 0.40 ng/mL, declining to 0.27 ng/mL (*p* < 0.0001) 48 h after cycle 1, whereas the high-ratio group (B) had a baseline of 0.23 ng/mL, increasing to 0.30 ng/mL (*p* < 0.0001). From the decrease in sTK1 among the patients with the higher baseline values, a half-life for TK1 of ~3.5 days can be estimated.

The two subgroups, defined by the sTK1 ratio post/pre-treatment cycle 1, showed a greater similarity during the following cycles. At cycle 2, the ratio had increased from 0.76 to 1.52 (*p* < 0.0001) in the low ratio group (A), whereas the high-ratio group (B) showed a similar value at cycle 2 (1.43) as at cycle 1 (1.38). At cycle 3, values had increased significantly in both groups, i.e., 1.88 in the low-ratio group A (*p* = 0.001) and 1.80 in the high-ratio group B (*p* = 0.04). At cycle 4, however, values did not differ from those at cycle 3; the values were 1.68 (*p* = 0.56) in the low-ratio group A and 2.06 (*p* = 0.46) in the high-ratio group B. The reliability of the data is supported by the normal distribution of the ratios over the total range, as shown in the total of 131 patients (Appendix A).

Considering the response to treatment at cycle 1, an additional 77 patients were analyzed. In this group, the pre-treatment median sTK1 value was 0.32 ng/mL. Forty-eight hours after cycle 1, the median value was similar, 0.33 ng/mL. Both these values are similar to those in the 54-patient subgroup (pre-treatment value 0.30 ng/mL, post-treatment value 0.28 ng/mL). The median of the sTK1 post/pre-treatment ratio, 1.12, was the same as for the 54-patient subgroup. In the 38 of the 77 additional patients who had a ratio of <1.12, sTK1 decreased from 0.35 to 0.27 ng/mL (*p* = 0.0004), whereas the 39 patients with a ratio of >1.12 showed an increase in sTK1 from 0.31 to 0.56 ng/mL (*p* < 0.0001).

For further analysis of possible causes of these two response patterns, as well as of their potential prognostic value, the two subgroups (54 and 77 patients), being similar with respect to baseline patient and tumor characteristics, were combined into a group of 131 patients (Table 2).

**Table 2 cancers-13-05442-t002:** Serum concentration of thymidine kinase 1(sTK1) pre- and 48 h post-treatment cycle 1, and sTK1 ratio post/pre-treatment cycle 1 in 54 women (Group 1) and 77 women (Group 2) and in the combined Group 1 + 2. The patients were subdivided by the median of the ratio post/pre-treatment. A—ratio < 1.12, B—ratio > 1.12. Median values (IQR).

A1	A2	A3	B1	B2	B3	
Before Treatment	48 h after Treatment	Ratio Post/Pre-Treatment	Before Treatment	48 h after Treatment	Ratio Post/Pre-Treatment	Statistics
Group 1 *n* = 54						
sTK1, ng/mL						
0.40(0.34–0.45)	0.27(0.2–0.37)	0.76(0.6–0.9)	0.23(0.17–0.30)	0.30(0.22–0.46)	1.38(1.18–1.82)	A1 vs. A2 *p* < 0.0001
(*n* = 27)			(*n* = 27)			B1 vs. B2 *p* < 0.0001
						A1 vs. B1 *p* < 0.0001
						A2 vs. B2 *p* = 0.139
Group 2 *n* = 77						A3 vs. B3 *p* < 0.0001
sTK1, ng/mL						
0.35(0.24–0.45)	0.27(0.17–034)	0.82(0.59–1.0)	0.31(0.21–0.44)	0.56(0.31–0.78)	1.60(1.32–2.29)	A1 vs. A2 *p* = 0.0006
(*n* = 38)			(*n* = 39)			B1 vs. B2 *p* = 0.0001
						A1 vs. B1 *p* = 0.495
						A2 vs. B2 *p* < 0.0001
Group 1 + 2 *n* = 131						A3 vs. B3 *p* < 0.0001
sTK1, ng/mL						
0.35(0.25–0.44)	0.27(0.19–0.34)	0.78(0.60–0.98)	0.26(0.19–0.42)	0.45(0.27–0.74)	1.46(1.29–1.98)	A1 vs. A2 *p* < 0.0001
(*n* = 65)			(*n* = 66)			B1 vs. B2 *p* < 0.0001
						A1 vs. B1 *p* = 0.007
						A2 vs. B2 *p* < 0.001
						A3 vs. B3 *p* < 0.0001

Individual ratios of all patients are depicted in Figure 4.

**Figure 4 cancers-13-05442-f004:**
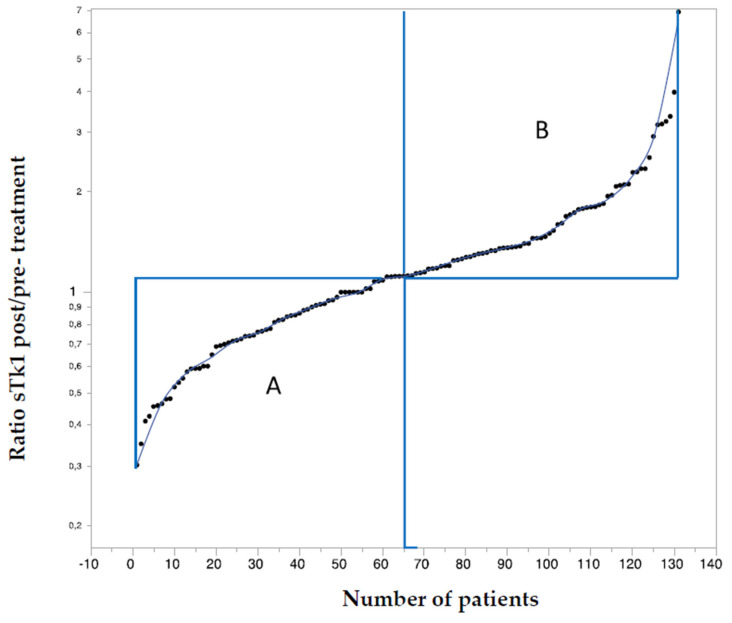
131 patients ranked according to the response ratio for treatment cycle 1.

The patients were divided by the median of the ratio (1.12) into two groups: A—64 patients with a ratio of <1.12 and B—65 patients with a ratio of >1.12. The reliability of the data is supported by the normal distribution of the ratios over the total range (Appendix A). In addition, in both groups, there was a significant linear relationship between pre-treatment values and values 48 h after treatment cycle 1: r = 0.73, *p* < 0.0001 (low-ratio group); r = 0.83, *p* < 0.0001 (high-ratio group) (Appendix A).

In search of associations between the different response patterns of sTK1 and patient and tumor properties, none of the baseline characteristics could be linked to the low- and high-ratio groups (Table 3).

With regard to the prediction of the pathologic response to NAC established at surgery after six treatment cycles, the frequency of local complete response and the distribution of stages in cases with remaining tumors were similar in the low- and high-ratio groups (Table 4).

A total of 26 and 19 patients, respectively, were node-negative (pN0), and the proportions of pN1 and >pN1 were similar.

Finally, the potential prognostic significance was studied by OS and DFS in the patients. After a median follow-up period of 60.3 months, 17 out of 65 (26.2%) of patients in the low-ratio group had died, as compared to 11 out of 66 (16.7%) in the high-ratio group; this difference is not significant. Thus, the sTK1 response per se does not appear to provide predictive or prognostic information.

Hence, we investigated the possible value of the sTK1 response to patients’ subgroup according to hormone receptor status. Out of 131 patients, 93 (71%) were ER+ and 36 (27.5%) ER- (receptor status was unknown in two cases). With regard to the prediction of pathologic response, in the 93 women with ER+ tumors, local complete response was reached in 13 patients in the low-ratio group and in 7 patients in the high-ratio group; correspondingly, remaining tumors were present in 37 and 36 patients, respectively (*p* = 0.25).

Survival curves for the ER+ patients, divided according to the sTK1 response into a low-ratio and high-ratio subgroup, are shown in Figure 5I.

After a median follow-up of 60.3 months, 11 out of 50 patients (20%) in the low-ratio group (B+) but only 1 out of 43 (2.3%) in the high-ratio group (A+) had died. This difference in mortality is statistically significant (*p* = 0.005). Among the ER- patients, there was no difference in survival between the low-ratio and high-ratio subgroups; out of 38 ER- patients, 16 (44.4%) had died. This percentage is significantly different from both the low-ratio (*p* = 0.01) and high-ratio (*p* < 0.0001) subgroups of the ER+ cases. DFS corresponded with OS (Figure 5II). Local recurrences and new contralateral recurrences appeared in 3 patients each and distant metastasis in 33.

Table 5 shows tumor characteristics according to the St. Gallen classification (21) for the low-ratio and high-ratio groups. A total of 47 tumors each were luminal-A- and luminal B-like, and 35 were triple-negative (1 ER+PR- T3 tumor, Ki67LI 80%, was classified as triple-negative, and 2 ER-PR+ T2/T3 tumors, Ki67LI 30% and 80%, were classified as luminal-B-like). It can be noticed that nearly equal proportions of luminal A and luminal B tumors were found in the A and B subgroups of ER+ tumors.

Table 6 shows the various baseline patient and tumor characteristics related to the St. Gallen classification [19] compared with the two sTK1 subgroups in ER+ tumor patients. None of the six significantly different tumor properties, which distinguished luminal A from luminal B tumors, were different in the sTK1 subgroups. Considering prognostic information, survival curves for luminal A and B cases are presented in Figure 5III,IV. In the present material, no significant difference in OS or DFS could be established between luminal A and luminal B tumors.

## 3. Discussion

In this study of BC, concentration changes of sTK1 during neoadjuvant cytotoxic treatment were assessed by serial measurements prior to and following 48 h treatment cycles 1–4. Repeated treatments enhanced the basic level of sTK1 before the cycles to double the baseline level, and 48 h after each treatment, sTK1 increased to about three times the baseline concentration. The exception was sTK1 48 h after treatment 1, which did not differ from the baseline level. Before treatment 2, however, sTK1 doubled.

When the dynamics of the individual reaction patterns was expressed as the post/pre-treatment sTK1 concentration ratio, patients could be subdivided according to their reaction at the first cycle into two groups of positive (ratio > 1.12) and negative (ratio < 1.12) responders. Unexpectedly, the sTK1 baseline level of the positive responders was significantly lower than that of the negative responders with higher baseline levels.

An inverse relationship between baseline sTK1 level and 48 h post-treatment response at cycle 1 was found in all tumor subgroups. No relationship was found between these response variables and any patient or tumor characteristics, including Ki67 LI, tumor volume, or hormone receptor status. Thus, this measure of tumor response provides information that cannot be obtained by established methods used for the subdivision of BC. When studying the prognostic value of the sTK1 response ratio in the group of patients as a whole, there was no association with the long-term outcome. However, if patients were sub-grouped into ER+ and ER-negative, a strong association was found between the response ratio and prognosis in ER+ patients, and the outcome was more favorable in patients with high response ratios. In the subgroup of ER-negative patients, on the other hand, the prognosis was independent of the response ratio. The affiliation of a patient with one of the two reaction groups was independent of the level of sTK1 at baseline and, in the high ratio group, of the level of increase over the baseline concentration 48 h after treatment. Thus, a tumor property, which is connected with response to chemotherapy and which provides prognostic information, is confined to a subgroup of patients with ER+ tumors. The dynamics of rapid shifts of sTK1 in individual patients was expressed by the ratio of two concentration values independently of other parameters, such as clinical tumor volume, which was used earlier in the assessment of cell loss [18]. As in the previous study, the association of the sTK1 reaction to chemotherapy with survival points to the tumor specificity of TK1.

The overall tumor reaction to chemotherapy at cycles 1 and 2 can be described in terms of an economic relationship between tumor and stromal cells. Stromal cells, which are responsible for the clearance of non-viable cells, are dimensioned in relation to the cell loss of the tumor. Cell loss in undisturbed tumor growth is in the order of 70% and 90% of the tumor cell birth rate, and the total number of cells lost has been estimated as 21 × 10^6^ and 487 × 10^6^ cells/day [20]. The first cycle of chemotherapy encounters a drug novice tumor, surrounded by tumor-associated normal cells. This system of normal cells is, however, sized only for cell clearance at a level of undisturbed tumor cell turnover. Thus, the system is unable to clear the sudden excess of disrupted cells generated by the first treatment cycle; this could explain the unchanged concentration of sTK1 48 h following treatment 1. To replenish phagocytic cells, approximately 2 days are needed for the evolution of monocytes in the bone marrow, and after about 1 day in the blood circulation, monocytes mature successively into macrophages and dendritic cells with lifespans of 4 and 7 days, respectively [21]. During the 3-week interval between cycles 1 and 2, the capacity of the clearance system reached a new, higher level which is reflected by the doubling of sTK1 just before treatment 2 and the three-fold sTK1 48 h following cycle 2. Taking into account tumor shrinking, the clearance capacity increased with treatment cycles 3 and 4. There was, however, no tendency towards any association between the response ratio at treatment cycles 2, 3, and 4 and long-term outcome such as that found at cycle 1. Thus, the differential reaction at cycle 1 was obviously obscured by the high inflow of sTK1 after treatment cycles 2, 3, and 4, although the post/pre-treatment sTK1 ratio at cycle 2 did not reach the ratio of about 2 found at cycles 3 and 4.

The sTK1 reaction to therapy at cycle 1 divided the patients into two groups with higher and lower baseline sTK1 levels. Variations in the level of sTK1 during treatment are more likely caused by changes in the rate of inflow of TK1 into the blood circulation rather than by changes in the clearance of TK1 from the blood by liver, spleen, and kidney. Hence, a higher concentration of sTK1 48 h after treatment 1 in the patients with the lower baseline sTK1 levels should reflect an increased inflow of TK1, possible due to cell death. On the other hand, lower concentrations of sTK1 48 h after treatment 1 than the baseline level found in the other group of patients should indicate a decreased inflow of TK1 into the circulation. Because a reduced cell death after chemotherapy is unlikely, possible reasons for a diminished inflow of TK1 into the blood could be due to a transient obstruction of the flow of cells through cell cycles, combined with sustained cell death, or due to increased clearance of disrupted tumor cells by mechanisms other than those resulting in the release of TK1 into the circulation. Besides the complicated interplay between the cell loss from the tumor and the normal environmental cells [22,23], which might determine the blood level of TK1, tumor aneuploidy cannot be excluded as a source of higher baseline levels of sTK1. Aneuploidy with increased amounts of DNA is found in about two-thirds of high stage BC [24] cases, and the corresponding higher amount of TK1 in aneuploid S-phase/G2 cells should also be reflected by increased blood levels of TK1 when these aneuploid cells die even if they are of lower proliferation.

Whatever the reasons for the poorer response to the first cycle of chemotherapy in the negative responders, it was possible to calculate a half-life of sTK1 of about 3.5 days from the difference between sTK1 at baseline and the 48 h concentration. Particularly, the assumed sudden arrest of inflow is unlikely, and therefore, the real half-life might be less than 3.5 days. This half-life is considerably shorter than that one of about 30 days assumed earlier [25,26,27]. Only a short half-life of blood biomarkers makes it possible to assess rapid changes of tissue reactions such as the cytotoxic cell death caused by repeated cycles of therapy.

ER+/HER2- tumors comprise about 80% of all newly detected BC [28]. This is a heterogeneous group of tumors in which the mainstays of prognosis are tumor size, grade, and nodal spread [29]. Patients in the present study were HER2- but otherwise non-selected, and 75% of the tumors were luminal-like. All patients were at high risk and received anthracycline–taxane-based NAC. For those patients, pCR is an established surrogate endpoint for survival [30]. In ER+ BC, however, pCR is reached in only <10% of patients and is not predictive in luminal-A-like tumors [31]. Studies on gene expression and acquired mutation in ER+ tumors have improved our understanding of response to endocrine therapy with the potential to predict outcomes in individual patients [32]. There is, however, a serious lack of predictive or prognostic biomarkers for cytotoxic treated patients able to guide therapy.

In the present study of 93 ER+/HER2- tumors, pCR (including yN0) was reached in 9/93 (9.7%) patients, and a total of 12/93 (12.9%) patients died during 60 months of follow-up. Out of 12 death cases, 1 (2.3%) belonged to the 43 patients of the high-ratio group (ratio > 1.12) and 11 (22%) to the 50 patients of the low-ratio group (ratio < 1.12). Out of 20 cases with recurrence, 4 (9.3%) were found in the high-ratio group and 16 (32%) in the low-ratio group.

The results of the study highlight the value of dynamic measurements during therapy that made it possible to identify a hitherto unknown tumor property, which is related to response to therapy. Because the tumors reacted to therapy as early as 48 h following the first treatment cycle, this tumor property was obviously existent before the treatment with epirubicin and docetaxel, which caused intercalation of DNA and RNA, and binding of microtubule, respectively [33]. This differential tumor reaction was found in all patients independent of the ER status of the tumors and before the patients received bevacizumab at cycles 3–6, which binds selectively to the vascular endothelial growth factor (VEGF). Differences in the outcome of the disease are, however, restricted to ER+ tumors. Therefore, this unknown tumor property might be connected via estrogen hormone receptors with genes involved in the regulation of cell proliferation [34,35] or mitotic checkpoint genes such as BubR1 [36]. The independence of the reaction of the lymph node status points to the significance of the early hematogenous metastatic spread of tumor cells, which is different from that to the regional lymph nodes. Early hematogenous metastatic spread was also stated in previous studies of lymph-node-negative and -positive BC patients with the 70-gene signature of the MammaPrint, which subdivides tumors into two risk groups [37].

In the same patients of the PROMIX trial, TK1 activity was also measured [38]. The mean TK1 activity was reported to increase from about 30 units at baseline to about 200 units 48 h following cycle 1. Such a 7-fold increase in activity was not observed for the protein content of TK1. Before treatments 2, 3, and 4, activity levels of about 900 units were found. This relative increase in activity was much higher than that of the protein content, whereas the 48 h post-treatment activity level of about 1100 units relative pre-treatment activity of cycles 2–4 was much lower than that of protein measurements. Possible reasons for these discrepancies might be related to differences in the performance of the methods in respect to the two molecular forms of thymidine kinases, TK1 and TK2 [39], various molecular active and inactive aggregates of TK1 in the serum [40,41], and the upregulation of TK1 in the repair of DNA damage following the cytotoxic insult of chemotherapy [14]. The TK1 activity measurements are derived from the incorporation of bromodeoxyuridine (a thymidine analogue) into DNA [42] and include the activity of both TK1 and TK2, while the protein measurements are based on two monoclonal antibodies against the C-terminal of the thymidine kinase 1 molecule. Before measurement of the TK1 protein content by the latter method, the serum sample is pre-treated with a buffer to break up high molecular weight complexes.

The possible influence of NAC on hormone receptors has been studied previously by comparing the hormone receptor status of the remaining tumor with that of the baseline. A shift from ER+ to ER-negative has been reported in 5–13% of tumors without significance for survival [43,44]. Early molecular subtype switches have been studied in needle biopsies taken 24–96 h after cycle 1 [45], and in another study of the PROMIX trial, after cycle 2 [46], during anthracycline–taxane-based NAC. In both studies, the most frequent change in molecular subtype was the early shift from luminal B to luminal A. This shift to luminal A has also been described previously, and possible mechanisms have been discussed, such as selective killing of higher proliferating, chemo-sensitive luminal B tumor parts, leaving the less responsive luminal A parts of the tumor, or retardation of the flow of cells through the cell cycle, rendering luminal B tumors more similar to luminal A tumors [47]. Decreased expression of cell cycle inhibitors early during NAC was associated with poor response and increased interferon signaling as well as high expression of cell proliferation genes in residual tumors were combined with the inferior outcome of disease [45]. In hormone receptor-positive tumors, only the baseline activity of the activated stoma signature was significantly associated with event-free survival [46]. In another study of the PROMIX trial, a decrease to low levels (<10%) of FOXP3+ T-lymphocytes in biopsies taken after cycle 2 of NAC was significantly associated with an improved DFS, whereas concentration changes of CD163 macrophages during NAC were unrelated to survival or tumor response [48].

This study has several limitations and should be considered to be exploratory. In the first part of the study, intended to obtain a survey of the reaction of sTK1 to exposure to a cytotoxic drug, the sample size was small. Fifty-four patients were selected for whom all eight serum samples of cycles 1–4 were available, thus avoiding any bias due to lacking values. The two different reaction patterns 48 h after the first cycle of NAC were, however, confirmed in serum from 77 additional patients collected some years earlier over the course of the trial. This and the fact that the samples were obtained from five hospitals in Sweden strongly contradicts the belief of an accidental effect of cytotoxicity. Association with tumor properties is based on survival studies in a subgroup of 93 ER+ tumor patients. Even this number is relatively small. The specificity and reliability of significant differences between the high- and low-ratio groups for OS as well as for DFS are supported by concomitant studies in ER- tumor patients without any difference in survival between the ratio groups. The response to therapy was studied after the simultaneously performed chemotherapy with epirubicin and docetaxel. Thus, the significance of intercalation of DNA by epirubicin and the effect of docetaxel on the microtubules per se could not be evaluated. The single-arm design of the PROMIX study did not provide any opportunity to evaluate the influence of bevacizumab on survival. Postoperative adjuvant radiotherapy and adjuvant endocrine therapy were given equally to all patients by the participating hospitals according to the Swedish guidelines and cannot explain the difference in the outcome of the two ER+ subgroups.

## 4. Materials and Methods

### 4.1. Study Design and Treatment

This study was part of the neoadjuvant, multicenter single-arm Phase II PROMIX trial. Trial Registration: PROMIX (Clinical Trials.gov NCT000957125).

The study was approved by the Ethics Committee at Karolinska University Hospital, 2007/1529–31/2. The inclusion criteria and treatment protocol are fully described elsewhere [46]. Briefly, between 2008 and 2011, 150 women with newly detected locally advanced but operable HER2- breast cancer with or without regional lymph node metastases were enrolled. Other inclusion criteria were age ≥ 18; adequate bone marrow; renal, hepatic, and cardiac functions; and no uncontrolled medical or psychiatric disorders. The main exclusion criteria were distant metastases, other malignancies, pregnancy, or lactation. Each patient fulfilling the criteria was given written and oral information about the study by the treating oncologist of the five participating centers. Written informed consent was obtained from all patients.

The patients were scheduled for 6 cycles of epirubicin and docetaxel (75 mg/m^2^ i.v. each) every 3 weeks, and in the absence of clinical complete response (cCR) after the second cycle, for the addition of bevacizumab (15 mg/kg i.v.) on day 1 of cycles 3–6. Within 3 weeks after completing chemotherapy, the patients underwent surgery, which included axillary lymph node dissection. Treatment following surgery was in accordance with the Swedish national guidelines [49]. The endpoint in this part of the PROMIX study was OS (time from surgery to death) and DFS (time from surgery to recurrence, metastasis, or death in cancer).

In the early course of the trial, blood samples were scheduled before and 48 h after treatment cycle 1 but only 48 h after treatment cycles 2–4. Later, a decision was made to collect blood samples also prior to cycles 2–4. As a consequence, samples for sTK1 measurements from all eight time points of the study were available in 54 patients, and before and 48 h after cycle 1 in 77 additional patients. Altogether, in 131 patients, sTK1 could be measured at baseline and 48 h after treatment 1.

### 4.2. Data Collection

Serum thymidine kinase 1 concentration: For collection of serum, venous blood was drawn in 5 mL plastic tubes. The tubes were inverted 10 times, and the blood sample was allowed to clot for 30–60 min and centrifuged for 10 min at 1500 RCF = g at room temperature. After transfer of serum to a new tube, it was centrifuged at 3000 RCF = g for 10 min at room temperature and transferred to new tubes in aliquots of 0.5 mL to be immediately frozen at −20 or −80 °C for storage at −80 °C until analysis.

Frozen serum was transferred to the Department of Anatomy, Physiology, and Biochemistry, Swedish University of Agricultural Sciences, Uppsala, Sweden, for measurement of the concentration of TK1 with the sandwich TK210 ELISA, produced by AroCell AB, Uppsala. This test is based on two monoclonal antibodies against the C-terminal region of the TK1 protein and was performed in duplicate according to the manufacturer’s instructions (for a detailed description of the laboratory procedures, see: www.e-labeling.eu/ARO1001–15-7, accessed on 5 September 2016). Samples were blinded with respect to patient identity, clinical data, and tumor pathology.

### 4.3. Tumor Pathology

Clinical stage was estimated from the largest diameter of the tumor at diagnosis by mammography and/or ultrasound. For calculation of volume (cm^3^), the tumors were considered to be spherical. Local pathologists of the five participating centers performed the histopathological analysis of tru-cut biopsies and immunohistochemistry in accordance with Swedish guidelines of pathology. ER/PR-positivity was defined as ≥10% positive stained tumor cells and a Ki67 LI of 20% for discrimination of luminal A from luminal B. The Swedish guidelines of pathology do not recommend grading of BC in core biopsies, and tumor grade was conducted only in 60% of the tumors. The tumors were classified according to the 2013 surrogate definitions of St. Gallen [19]. Pathologic complete response (pCR) was defined as absence of invasive cancer; residual non-invasive DCIS was allowed. Regional lymph node status was not taken into account for pCR. Remaining cancers were classified into pT1-pT3 according to their largest diameter.

### 4.4. Statistical Analyses

Associations between sTK1 and clinical data were evaluated with Fisher’s exact test, Wilcoxon’s test, and Spearman’s linear regression analysis. Data from serial measurements were evaluated by matched-pair analysis. Overall and progression-free survival was calculated with the Kaplan–Meier method, and differences in survival with the log-rank test. A *p*-value of 0.05% indicated statistical significance. Statistical analyses were performed using software JMP14.1.0, SAS, Cary, NC, USA.

## 5. Conclusions

Chemotherapy novice patients with BC reacted to docetaxel-based chemotherapy 48 h after the first treatment cycle either by an increase or decrease in the blood concentration of sTK1. These early changes in sTK1 made it possible to predict the outcome of disease in patients with ER+/HER2- cancer. If the findings can be confirmed, they could be used for stratifying chemotherapy in ER+ patients early during NAC. In addition, the reaction can be utilized to elucidate the mechanisms behind tumor sensitivity to chemotherapy before treatment, which would provide a new baseline biomarker in BC.

## Figures and Tables

**Figure 1 cancers-13-05442-f001:**
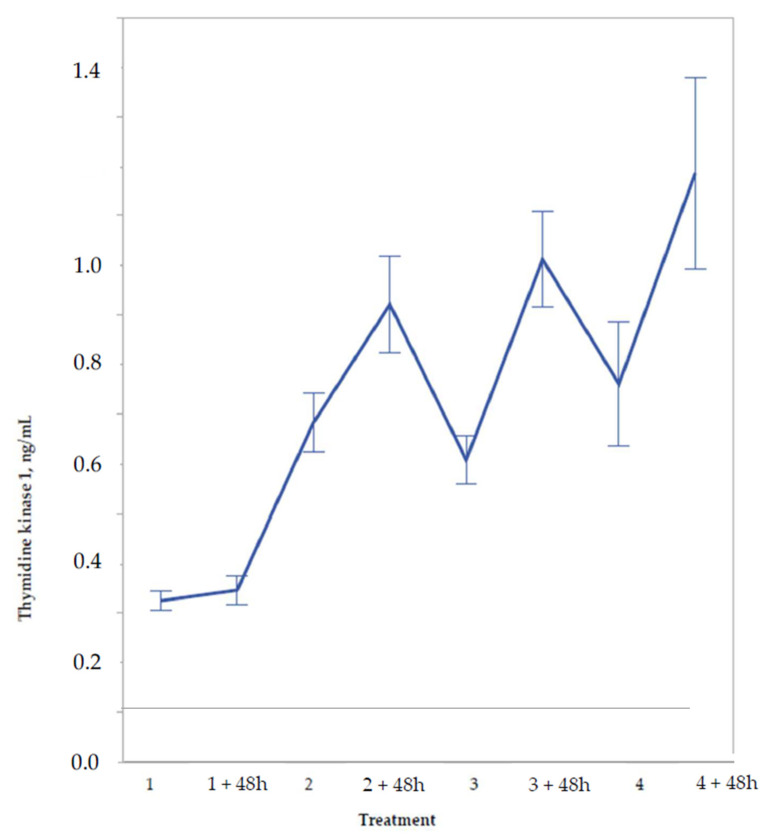
Serum concentration of thymidine kinase 1 in 54 patients before and 48 h after treatment cycle 1–4. Patients received treatment with docetaxel + epirubicin every 3 weeks. Mean values ± SE.

**Figure 2 cancers-13-05442-f002:**
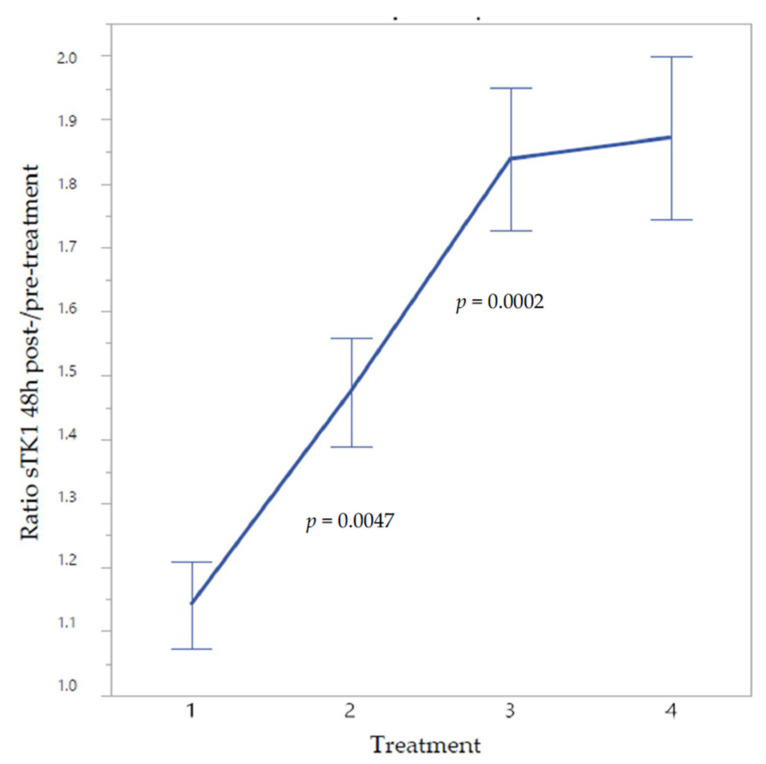
sTK1 response ratio 48 h post/pre-treatment at cycles 1–4 in 54 patients with localized breast cancer. The patients received neoadjuvant docetaxel + epirubicin every 3 weeks. Mean values ± SE.

**Figure 3 cancers-13-05442-f003:**
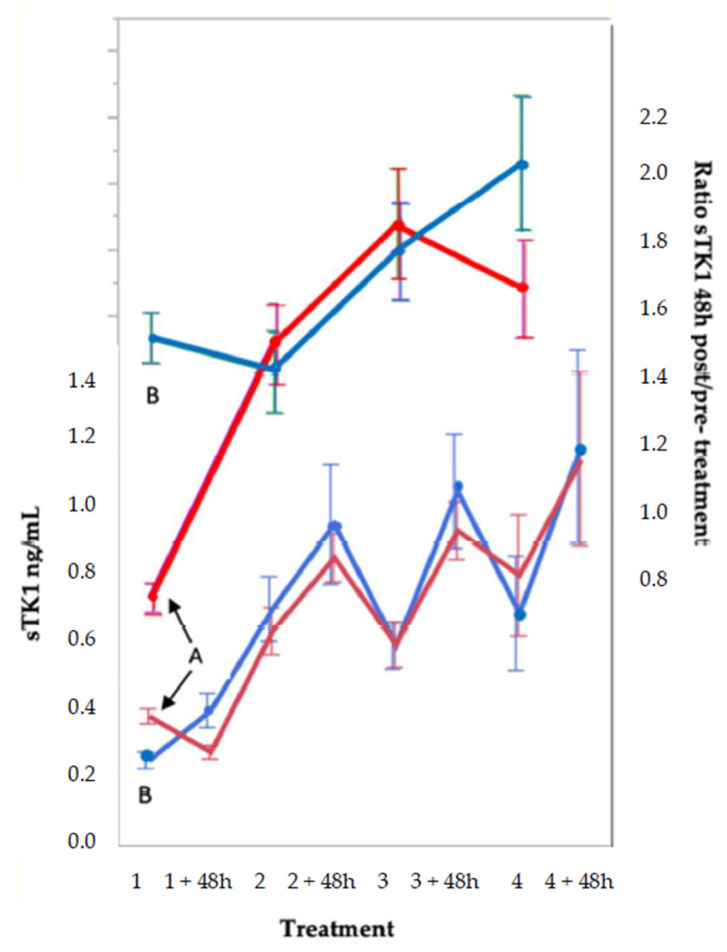
Serum thymidine kinase 1 concentration (sTK1) (lower part) and sTK1 ratio 48 h post/pre-treatment (upper part) at cycles 1–4 in 54 patients’ group. The patients have been divided according to the group median for the ratio by the median (1.12) of the sTK1 ratio 48 h post/pre-treatment at cycle 1 into 2 groups of 27 patients each. Mean values ± SE.

**Figure 5 cancers-13-05442-f005:**
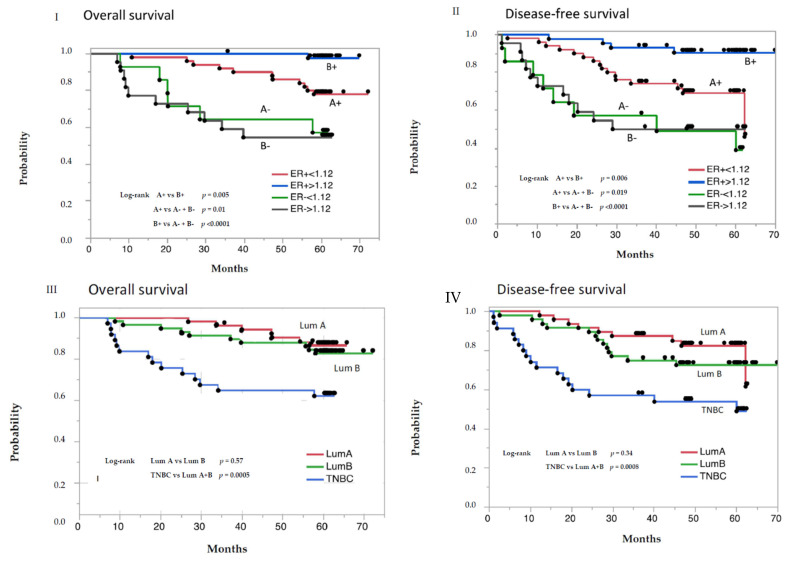
Overall survival (**left**) and disease-free survival (**right**) in 129 patients with breast cancer according to the ratio of serum thymidine 1 concentration 48 h post/pre-treatment at the first cycle of neoadjuvant chemotherapy and estrogen receptor (ER) status (**I**,**II**), and the 2013 St Gallen classification (**III**,**IV**). Abbreviations and number of patients in **I** and **II**: A+—50 ER+ patients, sTK1 post/pre-treatment ratio < 1.12; B+—43 ER+ patients, sTK1 post/pre-treatment ratio > 1.12; A-—14 ER- patients, sTK1 post/pre-treatment ratio < 1.12; B-—22 ER- patients, sTK1 post/pre-treatment ratio > 1.12. Abbreviations and number of patients in **III** and **IV**: Lum A—47 Luminal A patients, Lum B—47 Luminal B patients, TNBC—35 triple-negative patients.

**Table 1 cancers-13-05442-t001:** Baseline patient and tumor characteristics in 131 women.

Characteristics	Subgroup 1 (*n* = 54)	Subgroup 2 (*n* = 77)	*p* Value	Total (*n* = 131)
Age, years				
Median (range)	49.8 (33.7–70.6)	49.7 (30.0–69.2)	0.801	49.8 (30.0–70.6)
Menopausal status				
Pre-menopausal	30 (55.6%)	48 (61.3%)	0.187	78(59.5%)
Post-menopausal	24 (44.4%)	29 (37.7%)		53 (40.5%)
Stage				
1 (≤20 mm)	3 (5.5%)	1 (1.3%)	0.333	4 (3.1)
2 (>20 ≤50 mm)	19 (35.2%)	32 (41.6%)		51 (38.9%)
3 (>50 mm)	31 (57.4%)	40 (51.9%)		71 (54.2%)
Tx	1 (1.9%)	4 (5.2%)		5 (3.8%)
Tumor Volume, cm^3^				
Median (range)	87 (4.2–3052)	87 (4.2–904)	0.831	87 (4.23052)
Missing data	1	5		6
Grade				
1		4 (5.2%)	0.073	4(3.0%)
2	11 (20.4%)	29 (37.7%)		40 (30.5%)
3	14 (25.9%)	20 (26.9%)		34 (26.0%)
Missing data	29 (53.7%)	24 (31.2%)		53 (40.5%
Histological type				
Ductal	35 (64.8%)	56 (72.7%)	0.259	91 (69.5%)
Lobular	9 (16.7%)	11 (14.3%)		20 (15.3%9)
Other	8 (14.8%)	10 (13.0%)		18 (13.7%)
Missing data	2 (3.7%)			2 (1.5%)
ER status				
<10%	16 (29.6%)	20 (26.0%)	0.630	36 (27.5%)
>10%	37 (68.5%)	56 (72.7%)		93 (71.9%)
Missing data	1 (1.9%)	1 (1.3%)		2 (1.5%)
PR status				
<10%	20 (37.0%)	38 (49.4%)	0.167	58 (44.3%)
>10%	33 (61.1%)	38 (49.4%)		71 (54.2)
Missing data	1 (1.9%)	1 (1.2%)		2 (1.5%)
HER2				
2+	12 (22.2%)	19 (24.7%)	0.789	31 (23.7%)
0 or 1+	41 (75.9%)	58 (75.3%)		99 (75.6%)
Missing data	1 (1.9%)			1 (0.7%)
Nodal status				
Positive	32 (59.3%)	43 (55.8%)	0.697	75 (57.3%)
Negative	22 (40.7%)	34 (44.2%)		56 (42./%)
Intrinsic subtype				
Luminal A	22 (40.8%)	26 (33.8%)	0.699	48 (36.7%)
Luminal B	18 (33.3%)	30 (40.0%)		48 (36.6%)
TNBC	14 (25.9%)	21 (27.2%)		35 (26.7%)
Ki 67/Mib1, %				
Median (range)	30% (1–90%)	30% (3–90%)	0.282	30% (1–90%)
Missing data	3	6		9
Serum thymidine kinase 1,				
ng/mL				
Median (range)	0.30 (0.1–0.72)	0.32 (0.12–1.29)	0.487	0.31 (0.1–1.29)

Subgroup 1: in 54 women, pairs of serum for measurement of sTK1 were available prior to and 48 h following cycles 1–4. Subgroup 2: in 77 women, pairs of serum for measurement of sTK1 were available prior to and 48 h following cycle 1 only. Abbreviations: ER—estrogen receptor; PR—progesterone receptor; HER2—human epidermal growths factor receptor 2.

**Table 3 cancers-13-05442-t003:** Baseline characteristics in 131 women in groups A and B subdivided according to the post/pre ratio of thymidine kinase 1 concentration at the first cycle of chemotherapy.

Characteristics	A	B	*p*-Value
	Ratio < 1.12 (*n* = 65)	Ratio > 1.12 (*n* = 66)	
Age, years			
Median (range)	51.2 (30.0–69.2)	49.2(32.8–70.6)	0.58
Menopausal status			
Premenopausal	38 (58.5%)	40 (60.6%)	0.80
Postmenopausal	27 (41.5%)	26 (39.4%)	
Stage			
1 (≤20 mm)	4 (6.2%)	0	0.10
2 (>20 ≤50 mm)	23 (35.4%)	28 (42.4%)	
3 (>50 mm)	35 (53.8%)	36 (54.6%)	
Tx	3 (4.6%)	2 (3.0%)	
Tumor volume, cm^3^			
Median (range)	87 (4.2–3052)	87 (8.2–904)	0.61
Missing data	4	2	
Grade			
1	2 (3.1%)	2 (3.9%)	0.55
2	19 (29.2%)	21 (31,8%)	
3	14 (21.5%)	20 (30.4%)	
Missing data	30 (46.2)	23 (34.9(%)	
Histological type			
Ductal	45 (69.3%)	46 (69.7%)	0.28
Lobular	13 (20.0%)	7 (10.6%)	
Other	6 (9.2%)	12 (18.2%)	
Missing data	1 (1.5%)	1 (1.5%)	
ER status			
<10%	14 (21.6%)	22 (33.3%)	0.13
>10%	50 (76.9%)	43 (65.2%)	
Missing data	1 (1.5%)	1 (1.5%)	
PR status			
<10%	26 (40.0%)	32 (48.5%)	0.33
>10%	38 (58.5%)	33 (50%)	
Missing data	1 (1.5%)	1 (1.5%)	
HER2			
2+	14 (21.5%)	17 (25.8%)	0.60
0 or 1+	50 (76.9%)	49 (74.2%)	
Missing data	1 (1.5%)		
Nodal status			
Positive	36 (55.4%)	39 (59.1%)	0.69
Negative	29 (44.6%)	27 (40.9/%)	
Intrinsic subtype St Gallen			
Luminal A	27 (41.5%)	21 (31.8%)	0.34
Luminal B	24 (36.9%)	24 (36.4%)	
TNBC	14 (21.6%)	21 (31.8%)	
Ki 67/Mib1, %			
Median (range)	30% (1–90%)	30% (3–90%)	0.55
Missing data	6	3	
Serum thymidine kinase 1,			
ng/mL			
Median (range)	0.35 (0.12–1.29)	0.26 (0.1–0.71)	0.0068

Abbreviations: ER—estrogen receptor; PR—progesterone receptor; HER2—human epidermal growths factor receptor 2.

**Table 4 cancers-13-05442-t004:** Pathologic tumor response and axillary lymph nodes in 131 women after 6 treatment cycles with docetaxel + epirubicin. Women were subdivided into two groups A and B according to the median sTK1 ratio post/pre-treatment at cycle 1.

Stage/Lymph Nodes	Group A (*n* = 65)	Group B (*n* = 66)	*p*-Value
	Ratio < 1.12	Ratio > 1.12	
pT0	18 (27.7%)	18 (27.3%)	0.774
pT1	24 (36.9%)	22 (33.3%)	
pT2	13 (20.0%)	18 (27.3%)	
pT3	10 (15.4%)	8 (12.1%)	
pN0	26 (40.0%)	19 (28.8%)	0.610
pN1 (1–3)	23 (35.4%)	25 (37.9%)	
pN2 (4–9)	11 (16.9%)	13 (19.7%)	
PN3 (>9)	4 (6.2%)	8 (12.1%)	
pNX	1 (1.5%)	1 (1.5%)	

**Table 5 cancers-13-05442-t005:** Associations between thymidine-kinase-1-derived subtypes and intrinsic subtypes. A total of 129 women with breast cancer were subdivided according to the ratio sTK1 post/pre-treatment of cycle 1 and the estrogen receptor status. The four subgroups were compared with the St. Gallen intrinsic subtypes.

Intrinsic Subtype	Estrogen Receptor Positive		Estrogen Receptor Negative		Total
	Group A	Group B	Group A	Group B	
	Ratio < 1.12	Ratio > 1.12	Ratio < 1.12	Ratio > 1.12	
Luminal A	27 (54.0%)	20 (46.5%)	0	0	47
Luminal B	22 (44.0%)	23 (53.5%)	1 (7.2%)	1 (4.6%)	47
TNBC	1 (2.0%)	0	13 (92.8%)	21 (95.4%)	35
Total	50	43	14	22	129

**Table 6 cancers-13-05442-t006:** Baseline data of 92 patient with estrogen-receptor-positive breast cancer subdivided into Luminal A (*n* = 47), Luminal B (*n* = 45), and according to the reaction of sTK1 group A (*n* = 49) and B (*n* = 43).

Characteristics	Luminal A	Luminal B	*p*-Value	Group A	Group B	*p*-Value
	(*n* = 47)	(*n* = 45)		(*n* = 49)	(*n* = 43)	
Age, years				Ratio < 1.12	Ratio > 1.12	
Median (range)	49.4 (33.1–66.3)	53.4 (34.4–69.2)	0.262	52.8 (33.1–69.2)	49.2 (34.4–68.1)	0.194
Menopausal status						
Premenopausal	30 (63.8%)	22 (48.9%)	0.148	28 (57.1%)	24 (55.8%)	0.898
Postmenopausal	17 (36.2%)	23 (51.1%)		21 (42.9%)	19 (44.2%)	
Stage						
1 (≤20 mm)	3 (6.4%)		**0.030**	3 (6.1%)		0.192
2 (>20 ≤50 mm)	22 (46.8%)	15 (33.4%)		17 (34.7%)	20 (46. 5%)	
3 (>50 mm)	22 (46.8%)	28 (62.2%)		28 (57.1%)	22 (51.2%)	
Tx		2 (4.4%)		1 (2.1%)	1 (2.3%)	
Tumor volume, cm^3^						
Median (range)	65.4 (4.2–3052)	96.9 (14.1–1150)	**0.051**	96.9 (4.2–3052)	73.6 (8.2–904)	0.553
Missing data	1	2		2	1	
Grade						
1	3 (6.1%)	1 (2.2%)	**0.010**	2 (41%)	2 (4.6%)	0.898
2	21 (42.9%)	17 (37.8%)		19 (38.8%)	19 (44.2%)	
3	2 (4.1%)	12 (26.7%)		6 (12.2%)	8 (18.6%)	
Missing data	21 (42.9%)	15 (33.3%)		22 (44.9%)	14 (32.6%)	
Histological type						
Ductal	33 (70.3%)	32 (71.1%)	**0.037**	33 (67.4%)	32 (74.4%)	0.314
Lobular	12 (25.5%)	6 (13.3%)		12 (24.5%)	6 (14.0%)	
Other	1 (2.1%)	7 (15.6%)		3 (6.1%)	5 (11.6%)	
Missing data	1 (2.1%)			1 (2.0%)		
PR status						
<10%	5 (10.6%)	18 (40.0%)	**0.0009**	12 (24.5%)	11 (25.6%)	0.904
>10%	42 (89.4%)	27 (60.0%)		37 (75.5%)	32 (74.4%)	
HER2						
2+	12 (25.5%)	14 (31.1%)	0.507	13 (26.5%)	13 (30.2%)	0.740
0 or 1+	35 (74.5%)	30 (66.7%)		35 (71.5%)	30 (69.8%)	
Missing data	1 (2.2%)	1 (2.0%)				
Nodal status						
Positive	27 (57.4%)	24 (53.3%)	0.691	25 (51.0%)	26 (60.5%)	0.363
Negative	20 (42.6%)	21 (46.7%)		24 (49.0%)	17 (39.5%)	
Ki67/Mib 1, %						
Median, (range)	11% (1–30)	40% (20–90)	**<0.0001**	20% (1–90)	25% (3–85)	0.560
Missing data	2	2		4		
Thymidine kinase 1,						
ng/mL						
Median (range)	0.32 (0.13–1.29)	0.31 (0.1–0.93)	0.512	0.38 (0.12–1.29)	0.25 (0.1–0.68)	**0.001**

Bold indicates significance.

## Data Availability

Datasets used are available from the corresponding author on reasonable request.

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
