# Peer review of "Dynamics of Serum Thymidine Kinase 1 at the First Cycle of Neoadjuvant Chemotherapy Predicts Outcome of Disease in Estrogen-Receptor-Positive Breast Cancer"

_cancers, 2021, doi:10.3390/cancers13215442_

Round 1

Reviewer 1 Report

The significance of the results of this paper is very interesting. I just noticed very minor points. 

I can not find Figures S1, S2, and S3. Probably because I do not reach the website of this paper. 

In Figure 5, titles of survival curves are not complete and Roman numbers are not likely needed.

Author Response

We are grateful for the viewpoints, which we have found both of general interest and significant for improving the paper. The manuscript has been revised in accordance with all comments. We have also scrutinized the language of the manuscript and made several minor adjustments for the sake of clarity. We hope that the manuscript is now acceptable for publication.

Responses to Reviewer 1

Comment 1.

- In case there are still any problems of reaching the website with the re-submission, Figures S1, S2, and S3 are inserted below.

Comment 2.

- I agree. The titles of survival curves were completed, and roman numbers removed.

. The Roman numerals have been deleted.

Reviewer 2 Report

This is a well written and presented manuscript. A few errors are present (such as Table 3 typos). The conclusion presented is that serum thymidine kinase 1 predicts outcome of disease in ER positive breast cancer. While the data mostly supports this conclusion a few alterations are suggested and a few questions arise:

1) The prediction is for outcome in ER+ breast cancer is with regard to a particular therapy, not all therapies. Please make this clear in title and abstract in context of therapeutic setting.

2) This is a single author paper in which clinical staff, those who consented patients, those collected and analysed samples etc are not authors and mentioned only in acknowledgements. Can the author verify that those involved in the PROMIX group know that this manuscript is submitted? Can PROMIX be added to the abstract for search visibility?

3) The author acknowledges that the study numbers are low. Does the author have calculations to show the study is significantly powered to validate the statistical analysis shown?

4) The prediction, while correlating with outcome in ER positive cancer, is not high enough to base any clinical decision making upon. How does the author envisage the information could be used in a practical sense (applied) in a clinical setting? 

Author Response

We are grateful for the viewpoints, which we have found both of general interest and significant for improving the paper. The manuscript has been revised in accordance with all comments. We have also scrutinized the language of the manuscript and made several minor adjustments for the sake of clarity. We hope that the manuscript is now acceptable for publication.

Responses to Reviewer 2

Comment 1.

All tables were checked incl. Table 3 typos.

Comment 2.

In the Title as well as in Simple summary directed towards the non-specialist epirubicin/docetaxel was not mentioned. Such a detail, though important, could detract the interest from the subject. This is in line with the 4 quoted publications (ref.18, 19, 40, 49) of the PROMIX trial and the vast majority of publications on NAC. However, in Abstract,  epirubicin/docetaxel are mentioned.

One potential practical consequence of using early response markers would consist in a switch to another combination of cytotoxic drugs. This presupposes that tumour response, as well as the relationship between early response and long-term prognosis, may differ between treatment alternatives. As you suggested, in the revised manuscript, the specific therapy  is mentioned in  the abstract.

Comment 3

- Authorship and acknowledgment are in accordance with an agreement about the PROMIX study focussing primarily on how the effects of chemotherapy may be reflected in measures of sTK1, as obtained via the TK210 ELISA test. This includes analyses of how changes in sTK1 are related to pathologic response and long-term survival, identification of points in time where measures of sTK1 would be particularly informative, as well as how the predictive and prognostic value of sTK1 can be improved by combination with other patient and tumour characteristics.

I am fully aware that a study like the present could not be a one man’s work. The PROMIX trial has engaged physicians and other categories of health-care personnel in hospitals of different regions of Sweden. The collected data comprise not only potential markers for tumour response but also several established tumour and patient characteristics. Overall, the material constitutes a subject of numerous investigations, not only about thymidine kinase 1.

One principal criterion for co-authorship is the intellectual or scientific contribution that also implies that a co-author is capable of taking responsibility for the content. I highly esteem the work performed by clinicians, but their contribution, although invaluable, does not entail that they would be intellectually engaged in, or responsible for, every article that might emanate form the collected data. The same holds true for laboratory personnel if they use techniques which have been described in the literature or are commercially available.

The present study is based on raw data. No part of the analysis or text makes use of analyses or unpublished ideas obtained from the PROMIX group. The principal investigator and the study director of the PROMIX group have entrusted me the data and asked me to use the material in future investigations and to publish the articles as the single author. They are fully aware of the nature of my investigations. In the Abstract, the PROMIX trial as the source of the material as well as the Trial registration is named.

Comment 4

- Considerations of statistical power were part of the planning of the PROMIX study and included in the ethics application. The complexity of the underlying biological processes (e.g. factors and mechanisms determining tumour response to treatment and how tumour response might be reflected in various biomarkers) makes the exercise of mathematical modelling a very intricate task. Further, in the PROMIX trial, a set of patient and tumour characteristics, as well as several putative biomarkers were included. Such a strategy may provide not only the material for pre-determined specific analyses but it can also lead to unexpected insights or the recognition of tendencies motivating more specific investigations. Thus, regular power calculations could not be performed for every type of measurement.

In the post-hoc perspective, however, the issue of statistical power gains importance primarily in case of negative findings. For example, if the level of a biomarker has been measured before and after a treatment cycle, then an insignificant difference between group means would not justify the conclusion that the treatment does not have any influence on cellular mechanisms reflected by that biomarker. On the other hand, if a difference is statistically significant, this means, per definition, that the material was large enough. More philosophically, if an empirical finding stands in contradiction to a power analysis, made before a study, then, what should be rejected are the assumptions contained in the power analysis.

In studies where each test subject or patient serves as his or her own control, enabling paired comparisons or repeated-measures analyses, the inter-individual variability is less problematic than is studies where the effect of a treatment is evaluated by comparison with a control group.

Comment 5

- This is an interesting point. The present material may be considered small, as many evaluations of treatment regimens are based on thousands of patients. On the other hand, in clinical praxis the ideal test would be “significant” in the individual case. Thus, that the present method could reveal a relationship between early response and long-term prognosis in a relatively small group of patients makes it more likely an option in the clinical setting.

This is not to deny that the results should be confirmed in other groups of patients. But another task of follow-up studies would be to clarify how the methodology can be improved. One specific issue is the time course for changes in sTK1, induced by a treatment cycle, and the identification of a point in time when measurement of sTK1 would most likely be informative. Further, the precision in the method for measurement of sTK1 can be increased and the combination of sTK1 with other biomarkers would probably enable a more accurate characterisation of tumour chemosensitivity in the individual patient. The value of combining an unspecific marker like sTK1 with tumour-specific markers is highlighted by the present finding that the early sTK1 response had a prognostic value in the subgroup with ER-positive tumours but not in cases with ER-negative tumours.

However, to what extent methods for characterising tumours can be improved is an empirical question. The heterogeneity of malignancies, as well as imperfections in measurement procedures, suggests that there will never be an ideal set of biomarkers and treatment options enabling clinicians to make the best decision for every single patient. Nevertheless, biomarkers for tumour response would increase the factual basis for choosing the most appropriate treatment. Also, early identification of an ineffective treatment would reduce meaningless suffering.

I hope that numerous adjustments in the manuscript have made these issues clearer.

Reviewer 3 Report

The objective of the authors was to investigate whether alterations in serum concentration of thymidine kinase 1 (sTK1) levels during neoadjuvant chemotherapy (NAC) could be used as a biomarker to predict long term outcome in localized advanced ER+/HER- breast cancer patients. This study was part of neoadjuvant, multicenter single-arm Phase II PROMIX trial with OS and DFS as the end points (Clinical Trials.gov NCT 00957125). The authors have measured sTK1 levels before and 48h post cycle 1-4 in 54 patients and have measured sTK1 levels before and 48h post cycle 1 in 77 additional patient samples. The authors have reported increase in sTK1 levels 48h post treatment compared to the baseline levels. The authors have reported patients as positive vs negative responders based on their post/pre treatment sTK1 concentration ratio and have shown that the positive responders have significantly lower sTK1 baseline levels compared to negative responders. The authors have also reported strong association in response ratio and prognosis in ER+ patients but in ER- patients the prognosis is independent of response ration. The findings may need to be re-evaluated due to limitations of the study (which were detailed by the authors) but the reported findings are exploratory in nature and are interesting however the manuscript could not be considered for publication in the current form for following reasons.

Major concerns:

1) Did authors perform any mechanistic studies on the cell cycle dynamics in higher vs lower baseline sTK1 levels to further delineate the potential mechanism?

2) Could authors please explain the rational for analysis (or collection) of samples before and 48h post treatment in cycle 1-4 only instead of analyzing for cycles 1-6?

3) Did authors measure sTK1 levels in healthy subjects as well as part of this study?

4) Did authors analyze whether there are alterations in the levels of sTK1 in patients with and without regionally advanced BC?

Minor concerns:

1) Please correct the full name of "sTK1" in Ln # 137 of page # 5. It should have been "serum thymidine kinase 1" instead of "thymidine kinase". 

2) The number of patients by "grade" in the subgroup-1 were incorrect. The total patients are supposed to be "54" however the total number patients for all grades in subsection 1 are listed as "58". Could authors please look into it and fix the issue.

3) The number of patients in subgroup-2 are supposed to be 77 however the total number of patients by grade in subgroup-2 are mentioned as 73. Could authors please fix the issue.

4) Please correct the label in Fig 5C and Fig 5D from Lum to Lum A.

Author Response

We are grateful for the viewpoints, which we have found both of general interest and significant for improving the paper. The manuscript has been revised in accordance with all comments. We have also scrutinized the language of the manuscript and made several minor adjustments for the sake of clarity. We hope that the manuscript is now acceptable for publication.

Responses to Reviewer 3

Major concern 1

- This is a very interesting point. Since the sTK1 response to treatment 1 was related to long-term prognosis it apparently reveals properties of the tumour that are important and comparatively persistent, i.e. not radically altered by five cycles of chemotherapy.

The response patterns identified in the present study may constitute a clue as to factors and mechanisms determining the level of TK1 in blood. At the present stage, an overall model for sTK1 would not be realistic. Nevertheless, certain basic principles may be considered.

TK1 is released in conjunction with DNA synthesis. The mechanisms of release are unknown but release in cell death of malignant and normal tissues is obviously  different. Differences in release in conjunction with apoptotic cell death and necrotic cell death have been discussed.

 An additional factor could be the extent to which macromolecules from disrupted cells are recycled in the local microenvironment or released into blood. Earlier studies have shown that the contribution by normal tissues to the level of TK1 in blood is small.

The size of cell proliferation of the tumor and the volume of the tumor could be reflected by the level of sTK1 as shown in table 3 and mentioned in the discussion, there was no association between Ki67 LI and tumor volume. Furthermore, it is mentioned, how chemotherapy might influence the flow of cells through cell cycle and how aneuploidy could enhance the baseline level of TK1.

Major concern 2

- The rationale was to use all available and relevant data. The study concerns early response, which makes data from treatment 5 and 6 less relevant. The present analyses were based on all patients for whom there were complete data sets for treatment 1-4 (i.e. both pre-treatment and 48h values) and all for whom there were pre-treatment and 48h values for the 1st treatment.

As to the collection of data, during the first period of the PROMIX trial, pre-treatment and 48h samples were obtained only for treatment cycle 1. Then, a decision was made to collect paired samples also for the following treatments. A certain loss of data may also be due to circumstances of the clinical reality. For instance, a patient who does not feel well after a treatment may refrain from going to the hospital for leaving the 48h blood sample.

Major concern 3

- This study was based only on data from patients. Because of the inter-individual variability, among normal subjects as well as in patients, it is advantageous, in studies on the effect of chemotherapy, if every patient can function as his or her own control, as will naturally be the case if baseline values are available. Meaningful statistical analyses, with paired comparisons or repeated-measures analysis, can then be performed with a comparatively small number of patients, which is often not the case when comparing two different groups.

When it comes to the contribution to sTK1 by the untreated tumor, a matched control group with healthy women might have facilitated the interpretation of the inter-individual variability in the baseline values among the present patients. However, for the study of tumor response a repeated measures design is preferable.

Major concern 4

- The possible influence of regionally advanced BC on sTK1 has been analysed. No significant effect was found, neither at baseline, nor during treatment.

Minor concerns

- Thank you! Corrections have been done as suggested.

Round 2

Reviewer 2 Report

Figure 5 appears to have lost graph axis labeling formatting. 

I very much enjoyed the author's intellectual response to the questions and thank them for clarification and academic discussion of points. 

Reviewer 3 Report

The authors have addressed most of the concerns but have missed to address the minor concerns mentioned earlier. The manuscript could not be considered for publication in the current form as authors are requested to fix the following concerns. 

Minor concerns:

1) The authors did not fix the incorrect numbers listed to stratify patients by "grade" in the subgroup-1. The sum of all patients in Grade 1, 2, 3, and missing data groups should be 54 as per the authors however the total number of patients listed in the aforementioned categories comes down to 58. Could authors please cross check the numbers listed in Ln # 146 - 149 for subgroup-1 and fix the issue.

2) Similarly the authors did not fix the incorrect numbers listed for subgroup-2 by grade category. The number of patients in subgroup-2 are supposed to be 77 however the total number of patients by grade in subgroup-2 are mentioned as 73. Could authors cross check the numbers listed in Ln # 146 - 149 and fix the issue.

3) The labels on x-axis and y-axis for Figure 1, 3, and 5 are not properly aligned. Could authors please fix the issue.